# Acute physiological and perceptual responses to a netball specific training session in professional female netball players

Laurence P. Birdsey[1,2☉], Matthew Weston[3☉], Mark Russell[4☉], Michael Johnston[5☉], Christian J. Cook[6☉], Liam P. Kilduff[1,7☉]*

1 A-STEM College of Engineering, Swansea University, Swansea, United Kingdom, 2 English Institute of Sport, Manchester, United Kingdom, 3 Institute for Sport, Physical Education and Health Sciences, Moray House School of Education and Sport, The University of Edinburgh, Edinburgh, United Kingdom, 4 School of Social and Health Sciences, Leeds Trinity University, Leeds, United Kingdom, 5 British Athletics, Birmingham, United Kingdom, 6 Biomedical Discipline, School of Science and Technology, University of New England, Armidale, Australia, 7 Welsh Institute of Performance Science (WIPS), Swansea University, Swansea, United Kingdom

☉ These authors contributed equally to this work.

* L.kilduff@swansea.ac.uk

**Data Availability Statement:** Data are available from the Swansea University Institutional Ethics Committee (contact via coe-researchethics@swansea.ac.uk) for researchers

## Abstract

The 24 h responses to professional female netball-specific training were examined. British Superleague players (*n* = 14) undertook a 90-min on-court training session incorporating key movement, technical, and scenario-specific match-play drills. Perceptual (mood, fatigue, soreness), neuromuscular (countermovement jump peak power output [PPO], PPO relative to mass [PPOrel], jump height [JH]), endocrine (salivary cortisol [C], testosterone [T] concentrations) and biochemical (creatine kinase concentrations [CK]) markers were assessed at baseline (immediately before; Pre), and immediately, two and 24 hours after (+0h, +2h, +24h) training. Session (sRPE) and differential (dRPE) ratings of perceived exertion were recorded at +0h. Identification of clear between time-point differences were based on the 95% confidence interval (CI) for mean differences relative to baseline values not overlapping. At +0h, C (raw unit mean difference from baseline; 95% CI: 0.16; 0.06 to 0.25 µg·dl$^{-1}$), T (32; 20 to 45 pg·ml$^{-1}$), CK (39; 28 to 50 u·L$^{-1}$), PPOrel (2.4; 0.9 to 3.9 W·kg$^{-1}$) and PPO (169; 52 to 286 W) increased. At +2h, fatigue (15; 7 to 24 AU), CK (49; 38 to 60 u·L$^{-1}$), and soreness (14; 3 to 25 AU) increased, while T (-24; -37 to -11 pg·ml$^{-1}$) and mood (-20; -27 to -12 AU) reduced. At +24h, CK increased (25; 13 to 36 u·L$^{-1}$) whereas PPOrel (-1.6; -3.2 to -0.1 W·kg$^{-1}$) and JH (-0.02; -0.03 to -0.08 m) reduced. Responses were variable specific, and recovery of all variables did not occur within 24h. The residual effects of the prior stimulus should be accounted for in the planning of training for professional female netball players.

who meet the criteria for access to confidential data.

**Funding:** The funders had no role in study design, data collection and analysis, decision to publish, or preparation of the manuscript.

**Competing interests:** The authors have declared that no competing interests exist.

## Introduction

Netball is an intermittent team-sport with movement restrictions that yield unique activity and physiological demands that are position-specific [1, 2]. Players regularly perform a variety of on-court training sessions, including game-based (aimed at replicating match-play movement demands, technical skills and decision making under pressure) and skills-based (aimed at developing technical skills including passing, catching, shooting and movement patterns) sessions in addition to on and off-court conditioning [3], four to eight times per week. Additionally, at specific points in the season, competitive teams may play two matches per week, whilst performing regular training involving a mixture of components in order to maintain or improve specific fitness, skills and to refine match tactics [1]. In order to adapt to such stimuli, players must be able to perform to a high standard and recover for subsequent training or competition [4]. However, whilst the external load of elite netball match-play has been reported [1, 2], in addition to internal and external load of training [3, 5], limited data has profiled the fatigue and/or recovery associated with netball-specific training [6].

While netball-specific training responses are scarce, the acute post-exercise responses to training in other sports have been extensively reported following isolated strength [7], endurance [8] and soccer [9] training, with a single observation following speed training [10]; all of which have application to the demands of netball players. However, as players perform training to improve aspects related specifically to match performance, sport-specific training sessions are key to fully understand the responses of netball-specific training. Following soccer team-sport training, immediate increases in testosterone and decreases in cortisol concentrations have been observed in addition to a bi-modal recovery pattern of neuromuscular performance, with an initial decrease immediately post, partial recovery at two, and further impairment at 24 h post [9]. However, in female players, a delayed endocrine response has been reported of 24 h, with responses evident up to 72 h post-training [11], whilst following field hockey training, exercise intensity influences the endocrine response [12]. A greater understanding of the acute responses to, and recovery profile from on-court netball training may assist coaches and conditioning coaches to effectively plan the content of individual sessions, as well as the positioning of training within the week.

At present, there are limited reports upon the acute responses to team-sport training in females, with only one report following netball-specific training [6]. Knowledge of both the training stimulus, as well as the recovery response are necessary to prevent cumulative fatigue [6, 13] and allow recovery for adaptation. It is therefore imperative that coaching and conditioning staff have an understanding of the acute responses to specific training sessions to assist with effectively planning and optimising training. The purpose of this study was to characterise the neuromuscular, physiological, biochemical, endocrine and perceptual responses over a 24 h period to a netball specific training session performed by professional female netball players.

## Materials and methods

### Participants

Fourteen female netball players (age: 23 ± 4 years, mass: 73.2 ± 8.0 kg, height: 1.8 ± 0.1 m) from a British domestic Superleague team (representing the highest tier of professional netball in the UK) were recruited for this study that was conducted in December during the 2016 pre-season period (after a two-month period requiring strength, speed, endurance and netball-specific training sessions for four to six times a week). Players were included as members of this professional netball team and determined to be available for training by the team physiotherapist. No information was gathered regarding hormonal contraceptive use or menstrual cycle

phase, and the variance in basal testosterone concentrations possibly reflects either, or both, of these factors. This study was approved by the Swansea University ethics committee and players were informed of the benefits and risks of the investigation prior to signing an approved informed consent document and health screening questionnaire and were made aware that all material would be anonymised. All mandatory health and safety procedures were complied with in completing this research study.

## Design

This observational study was conducted over a 24 h period that followed an on-court netball-specific training session commencing at 16:00 h. Immediately prior to the training session baseline (Pre) samples of whole blood (creatine kinase concentrations; CK) and saliva (cortisol; C, and testosterone: T concentrations) were collected, countermovement jump (peak power output; PPO, PPO relative to mass; PPOrel, jump height; JH) testing was performed (preceded by a standardised warm-up), and perceived mood (adapted brief assessment of mood questionnaire: BAM+; [14]) was recorded. Immediately post-training (+0h), the above measures were conducted, however countermovement jump testing was performed within five minutes of the end of the training session. Session ratings of perceived exertion (sRPE; [15]) together with differential rating of perceived exertion (dRPE; [16]) were taken instead of the BAM + immediately following the countermovement jump testing at +0h. Exercise intensity was quantified by external (accelerometry) and internal (heart rate; HR, sRPE and dRPE) load metrics. Measures recorded immediately prior to the training session were repeated two (+2h) and 24 h (+24h) post-training.

All players were prescribed a light conditioning training session the day before testing, with the testing day being the second training session of the week. In preparation for training, players were instructed to eat and drink as usual (i.e. a high carbohydrate meal to support carbohydrate availability for the training session) and consumed a standardised meal prescribed by the team nutritionist to support recovery (i.e. high in carbohydrates to replenish carbohydrate stores, in protein to support muscle protein resynthesis, and with fruit and vegetables as part of a balanced diet) immediately following the measurements collected post-session at +0h. Thereafter, players were instructed not to perform any further structured exercise following testing. The next day, 24 h post-training, players reported for follow-up testing (i.e., +24h) having prepared nutritionally as if they were attending another training session. Due to players training schedules, +24h was the final available time-point before players performed a subsequent training session.

## Netball-training session

The training session performed by players was 90 minutes in duration and took place entirely on-court, commencing at 16:30 h. This was a routinely performed training session by the team with the aim of developing or maintaining technical skills, movement patterns, physical conditioning, match tactics and decision making under pressure, whilst replicating the intensity experienced in match-play. Players performed a warm-up of approximately 20 minutes consisting of a team exercise involving short intermittent sprints, dynamic stretching, ball skills and netball specific attacking and defending movements. Players then performed an exercise involving defenders aiming to intercept passes from the players in possession with the aim of developing decision making and technical skills under pressure. The training session progressed to the performance of scenario-specific match-play with three games lasting five to eight minutes interspersed with four to 11 minutes of recovery. This was performed by all

players, aimed to replicate match-play intensity, involved specific scenarios aiming to develop key areas of netball performance and had been regularly performed by players.

## Mood

Players recorded perceived mood using a modified version of the brief assessment of mood (BAM+; [14]). Using a bespoke application on an Android tablet (Iconia One 7 B1-750, Taipei, Taiwan: Acer inc), a series of 10 questions were answered one at a time with a 100 mm visual analogue scale anchored with "not at all" and "extremely" to record how players felt at that moment in time. The questions assessed: alertness, sleep quality, confidence, motivation, anger, confusion, tension, depression, fatigue and muscle soreness. Individual values for perceptions of fatigue and muscle soreness were assessed, in addition to an overall mood score, generated by subtracting the mean score of negative related items from the mean score of the positively related questions using the equation below [14]:

Mood score = (alertness + sleep quality + confidence + motivation) / 4 - (anger + confusion + tension + depression + fatigue + muscle soreness) / 6.

This method of calculating mood using the BAM+ has been reported to have acceptable internal consistency (Cronbach alpha score of 0.65 to 0.82; [14]), to be moderately correlated to high intensity match activity (measured by global positioning system) and is sensitive to physiological responses to competition in elite team-sport athletes [14] including netball [2].

## Endocrine function

For salivary hormone analysis, two ml of saliva was collected via passive drool [17] into sterile containers, with participants instructed to avoid eating food and drinking fluids other than water for 60 minutes prior to sampling to avoid contamination of samples. Samples were stored at -70˚C until assay, when, after thawing and centrifugation (2000 revolutions·min$^{-1}$ for 10 minutes), the saliva samples were analysed in duplicate for testosterone and cortisol concentrations using commercially available kits (Salimetrics, LLC, State College, PA, USA). The minimum detection limit for the testosterone assay was 6.1 pg·ml$^{-1}$, with interassay coefficient of variation (CV) of < 10%. The cortisol assay had a detection limit of 0.12 ng·ml$^{-1}$ with interassay CV of < 7%. Samples for each participant were assayed in the same plate to eliminate inter-assay variability.

## Creatine kinase activity

Whole blood capillary samples (120 μl) were collected from the fingertip and stored on ice in EDTA prepared collection tubes (Microvette 500, Sarstedt, Numbrecht, Germany) before centrifugation (3000 revolutions·min$^{-1}$ for 10 minutes). Plasma samples were stored at −70˚C before being analysed for CK activity using commercial kits (CK-NAC, ABX Diagnostics, Northampton, United Kingdom) on a spectrophotometer (Cobas Mira; ABX Diagnostics, Northampton, United Kingdom). Samples were measured in duplicate (CV = 3%) and recorded as a mean.

## Neuromuscular performance

A portable force platform with built-in charge amplifier (type 92866AA, Kistler Instruments Ltd., Farnborough, UK) was used to measure ground reaction force time history of countermovement jumps. A sample rate of 1000 Hz was used for all jumps and the platform's calibration was confirmed pre-testing. Power (CV = 2.4%) and JH calculated from takeoff velocity (CV = 3.4%) were calculated using standard procedures established in previous investigations

[10, 18]. These measures were chosen as they are sensitive to changes following team sport training [9], netball training [6] and netball match-play [2], and can be considered most sensitive to fatigue in the early stages of the recovery process [19]. Participants performed a standardised warm-up before jumping at all time-points, apart from +0h when players had immediately finished the training session, and performed two jumps, with the best jump used in subsequent analyses. All players were familiarised with this testing procedure as part of routine performance monitoring, were instructed to jump as fast and as high as possible and to keep hands on hips throughout the jump.

## Exercise intensity

Activity during the training session was recorded using commercially available units (Catapult S5, Catapult Innovations, Leeds, UK) housing a tri-axial accelerometer sampling at a rate of 100 Hz. To minimise movement artefacts, participants wore a custom-made vest (Catapult Innovations, Leeds, UK) in which the units were held in place vertically on the upper back. Data was downloaded using the manufacturer's software (Catapult sprint 5.1, Catapult Innovations, Leeds, UK) and analysed for external load (represented as Player Load™) with detailed calculations described previously [20]. This marker of intensity has been reported to be a valid and reliable method [20, 21] of measuring activities performed in team-sports movements (CV = 1.9%; [20]) and has been widely used in team-sports including netball [1, 2]. Participants wore a heart rate monitor (Polar Team System 2, Polar Electro, Warwick, UK) throughout the training session, recorded at beat to beat intervals, with data downloaded and analysed retrospectively using the manufacturer's software (Polar Team 2, Polar Electro, Warwick, UK). Data were analysed for external load (AU), external load intensity (AU·min$^{-1}$) and mean heart rate for the entire session, the entire session excluding breaks between drills (e.g., active periods only, excluding coaching interactions and recovery periods), and for the match-play portion excluding breaks between games.

## Ratings of perceived exertion

Immediately following the training session players recorded sRPE along with dRPE for breathlessness (RPE-B), leg muscle exertion (RPE-L), upper body muscle exertion (RPE-U) and cognitive/technical demands (RPE-T). Using a bespoke application on an Android tablet (Iconia One 7 B1-750, Taipei, Taiwan: Acer Inc.) ratings were provided using a numerically blinded CR100® scale with verbal anchors. dRPE provides a detailed quantification of internal load during team-sport activities [22], is a sensitive marker of match exertion [16] and distinguishes between different areas of effort [16, 22].

## Statistical analyses

We elected not to perform a power calculation as they are of little value in early exploratory studies, such as ours, where scarce data are available on which to base the calculations [23]. Data were analysed via a mixed effects linear model (SPSS v.21, Armonk, NY: IBM Corp.). Fixed effects in the model were time (Pre, +0h, +2h, +24h), with a random effect for player to account for the repeated measures nature of the study design. Uncertainty in our estimates is presented as 95% confidence intervals. Effects are presented as simple effect sizes (mean differences in raw units) which are independent of variance and scaled in the original units of analysis [24], thereby maximising the practical context of findings [25]. Our interpretation of between-time point differences in all dependent variables was based on the width of the respective 95% confidence intervals for the mean difference, with no overlap of the confidence intervals being a clear difference. We have presented, but not interpreted, standard effect sizes

(mean difference/pooled standard deviation of the Pre time point; SD) and percentage change scores. Further, a region denoting 0.2*SD, commonly referred to as a smallest worthwhile effect, has been included in all forest plots (see Figs 1–3).

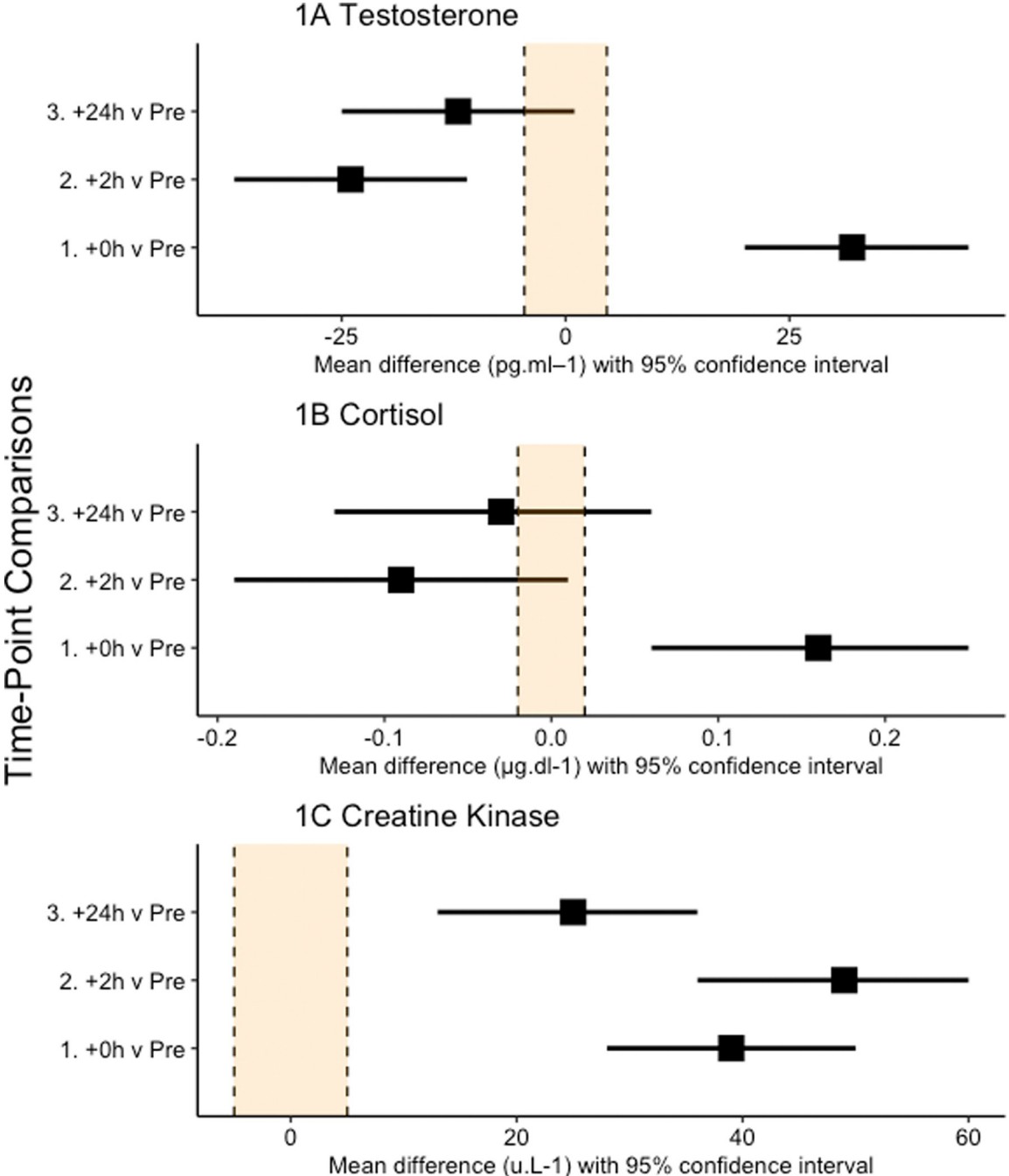

**Fig 1.** Effect statistics (mean difference and 95% confidence intervals) for the comparison of testosterone (A), cortisol (B) and creatine kinase (C) concentrations immediately (+0h), two (+2h) and 24 (+24h) hours following the performance of the training session compared to baseline. Zero (0) on the axis represents no difference between that time-point and baseline.

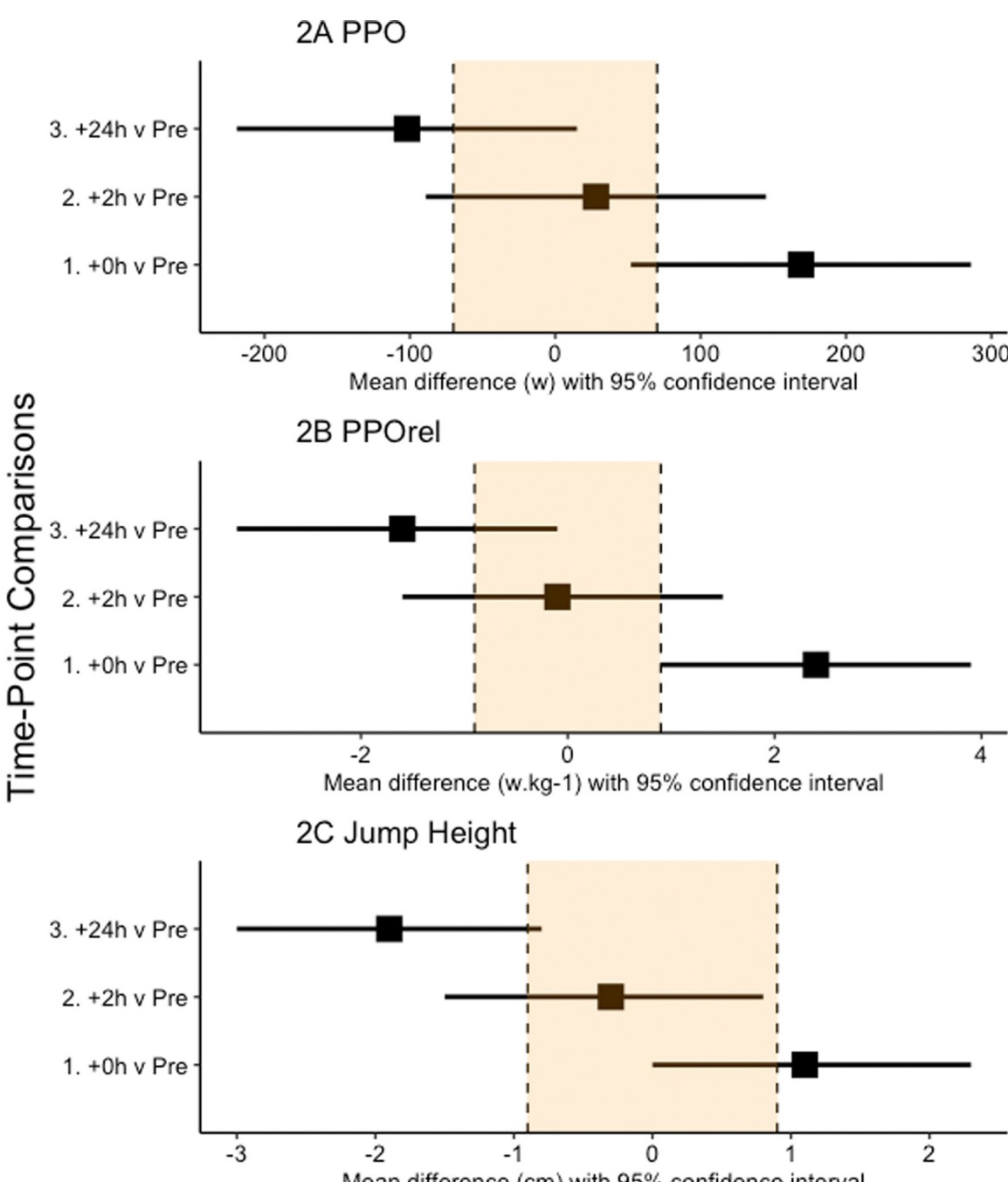

**Fig 2.** Effect statistics (mean difference and 95% confidence intervals) for the comparison of peak power output (A), peak power output relative to mass (B) and jump height (C) immediately (+0h), two (+2h) and 24 (+24h) hours following the performance of the training session compared to baseline. Zero (0) on the axis represents no difference between that time-point and baseline.

## Results

Descriptive training data are presented in Table 1. When compared to Pre, there was a clear increase in testosterone concentrations at +0h, followed by a clear decrease at +2h, but no difference at +24h (Fig 1A). The standardised effect sizes (% changes) for the comparisons were 1.19 (+42%), -1.30 (-31%) and -0.58 (-16%). For cortisol concentrations there was a clear

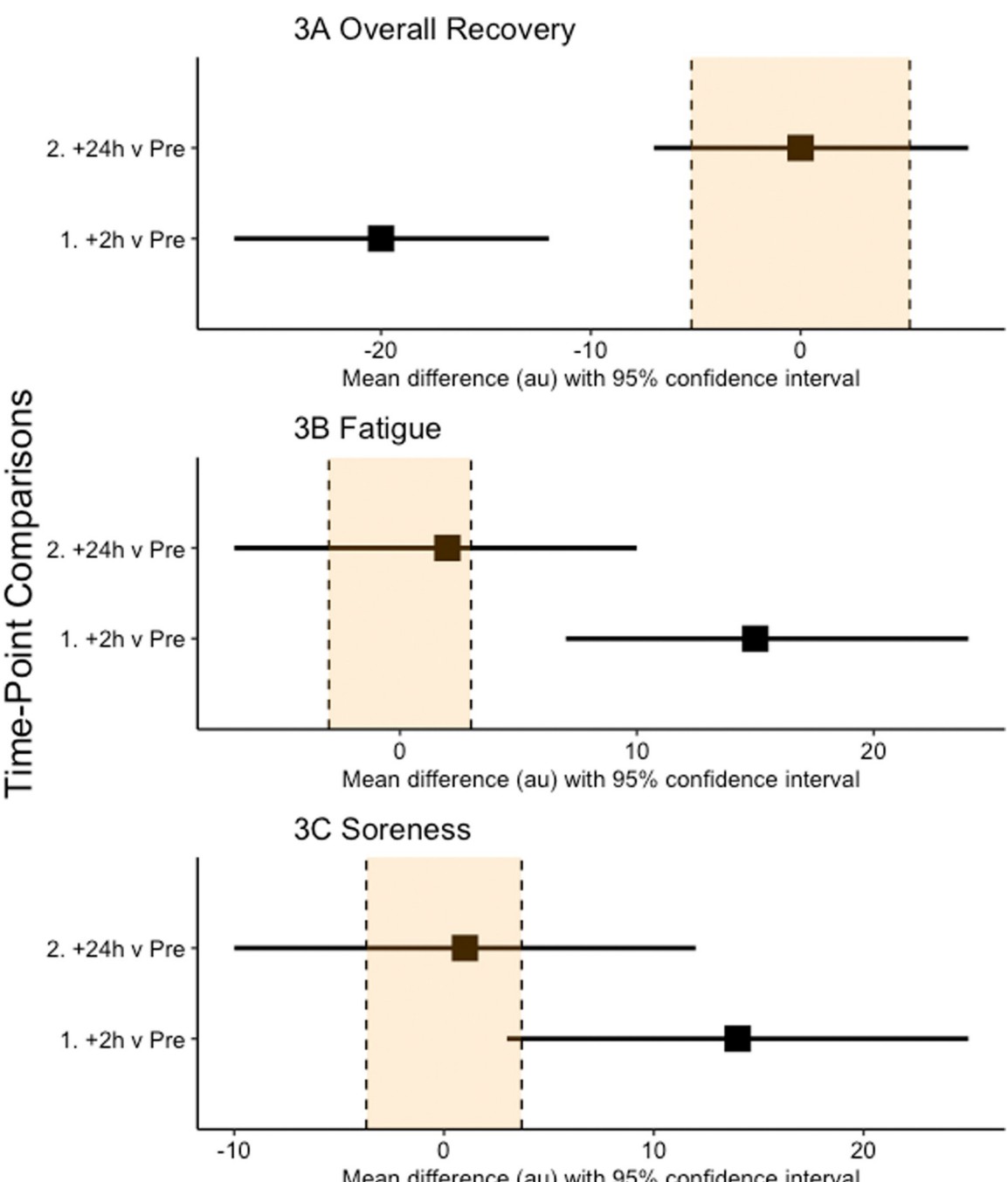

**Fig 3.** Effect statistics (mean difference and 95% confidence intervals) for the comparison of overall recovery (A), fatigue (B) and soreness (C) immediately (+0h), two (+2h) and 24 (+24h) hours following the performance of the training session compared to baseline. Zero (0) on the axis represents no difference between that time-point and baseline.

increase at +0h but no difference at +2h or +24h (Fig 1B). The standardised effect sizes (% changes) for the comparisons were 0.95 (+70%), -1.24 (-40%) and -0.39 (-15%). For creatine kinase concentrations there was a clear increase at all time points (Fig 1C). The standardised effect sizes (% changes) for the comparisons were 1.64 (+31%), 1.87 (+38%) and 0.98 (+17%).

**Table 1. Mean (± SD) of average HR, maximum HR, external load and external load intensity of the entire training session, the active and the match-play portions, and sRPE and dRPE for the entire netball training session.**

|  | Mean (± SD) |
| --- | --- |
| Mean HR (b·min$^{-1}$) | 147 (± 13) |
| Mean active HR (b·min$^{-1}$) | 167 (± 12) |
| Mean match-play HR (b·min$^{-1}$) | 171 (± 9) |
| Maximum HR (b·min$^{-1}$) | 192 (± 10) |
| Maximum match-play HR (b·min$^{-1}$) | 189 (± 9) |
| Total external load (AU) | 513 (± 81) |
| Active external load (AU) | 482 (± 78) |
| Match-play external load (AU) | 173 (± 35) |
| Total external intensity (AU·min$^{-1}$) | 5.6 (± 0.9) |
| Active external intensity (AU·min$^{-1}$) | 9.0 (± 1.5) |
| Match-play external intensity (AU·min$^{-1}$) | 8.1 (± 1.7) |
| sRPE (AU) | 74 (± 22) |
| RPE-B (AU) | 68 (± 24) |
| RPE-L (AU) | 62 (± 27) |
| RPE-U (AU) | 37 (± 21) |
| RPE-T (AU) | 63 (± 25) |

*Abbreviations*: SD: standard deviation; HR: heart rate; AU: arbitrary unit; sRPE: session rating of perceived exertion; dRPE: differential rating of perceived exertion; RPE-B: rating of perceived breathlessness; RPE-L: rating of perceived leg muscle exertion; RPE-U: rating of perceived upper body muscle exertion; RPE-T: rating of perceived cognitive/technical demand.

When compared to Pre, there was a clear increase in PPO at +0h but no difference at +2h and +24h (Fig 2A). The standardised effect sizes (% changes) for the comparisons were 0.47 (+5%), 0.07 (1%) and -0.27 (-3%). For PPOrel, there was a clear increase at +0h, no difference at +2h followed by a clear decrease at +24h (Fig 2B). The standardised effect sizes (% changes) for the comparisons were 0.50 (+5%), -0.02 (0%) and -0.34 (-3%). For JH, there was no difference at +0h or at +2h, but a clear decrease at +24h (Fig 2C). The standardised effect sizes (% changes) for the comparisons were 0.25 (+4%), -0.07 (-1%) and -0.39 (-6%).

When compared to Pre, there was a clear decrease in overall mood score at +2h but no difference at +24h (Fig 3A). The standardised effect sizes (changes in raw units) for the comparisons were -0.84 (-20 AU) and 0.01 (+0 AU). For fatigue, there was a clear increase at +2h, but no difference at +24h (Fig 3B). The standardised effect sizes (changes in raw units) for the comparisons were 1.01 (+15 AU) and 0.13 (+2 AU). For soreness, there was a clear increase at +2h, but no difference at +24h (Fig 3C). The standardised effect sizes (changes in raw units) for the comparisons were 0.80 (+14 AU) and 0.06 (+1 AU).

## Discussion

The aim of this study was to characterise the neuromuscular, physiological, biochemical, endocrine and perceptual responses (over 24 h) to a regularly performed netball-specific training session in professional female players. The primary findings highlighted that responses over 24 h differed according to the marker being examined. Markers were elevated at 2 h post-exercise and a return to baseline was not achieved 24 h post-training for all variables examined. Accordingly, these data indicate that the residual effects of the previous training bout should be considered when planning subsequent training in the 24 h following a netball training

session; findings which will likely be of interest to staff involved in the planning and periodisation of training for female netball players.

Immediately post-training, neuromuscular performance was increased in addition to higher testosterone and cortisol concentrations when compared to baseline. Hormonal responses to exercise can be influenced by training intensity, whilst the increase observed in the present study is similar to that observed after high intensity hockey training [12], technical netball training [6] and following elite netball competitive match-play [26]. Increases in testosterone and cortisol concentrations could be attributed to an increase in competitiveness and dominance behaviours [26, 27], as well as energy provision and muscle tissue repair following exercise-induced muscle damage [28]. Mean basal testosterone concentration in the present study was high (mean ± SD; 77.4 ± 23.0 pg·ml$^{-1}$) in relation to a non-elite female population [29], but were in line with previous reports in elite female athletes [29, 30] and international netball players [6]. Additionally, as no control was in place for menstrual cycle phase or hormonal contraceptive use, exercise responses and recovery patterns may be influenced according to these factors [12]. Exercise-related increases in CK concentrations, proposed to be indicative of skeletal muscle damage [31], have been reported to be associated with impaired neuromuscular function [32]. However, the findings of the present study show elevated neuromuscular performance in spite of an increase in CK, similar to previous reports following Women's rugby sevens [33]; questioning the relationship between CK and neuromuscular performance.

Following an exercise stimulus mechanisms of both fatigue and muscle potentiation coexist, with the resulting performance benefit dependent upon the balance of these two factors [34]. The increase in testosterone concentration at +0h observed in the present study may have positively influenced behaviour, contractile signalling and performance [35], and subsequent positive influence on neuromuscular function to a greater extent than impairment through muscle damage or fatigue. Additionally, muscle temperature may have increased following the training session, along with induction of post-activation potentiation due to dynamic movements [36], greater than achieved following the standardised warm-up.

Two hours following the training session, cortisol concentrations returned to baseline, whilst testosterone concentrations reduced below baseline values, similar to that following a technical netball training session [6]. Although previous reports following other activities highlight a varied acute testosterone and cortisol response following a variety of sport and exercise stimuli [7, 8, 10, 37], the decrease in testosterone concentrations in the present study may be associated with circadian rhythm changes as previously reported [38]. Cortisol concentrations could therefore be considered elevated in relation to the expected circadian rhythm response, which could highlight an increased catabolic state at this time-point. Additionally, CK concentrations were elevated, and mood state negatively affected compared with baseline. Findings suggest, that if multiple training sessions are to be performed on the same day, as is often performed by team-sport players, including netball, then more than two hours should be provided to allow sufficient recovery of perceptual markers for subsequent performance.

Whilst most variables recovered to baseline 24 h post-training, CK remained elevated, and markers of neuromuscular performance remained suppressed. As players performed a subsequent training session at this time-point as part of their team's elite training schedule, no further measures could be obtained. Perceptual markers of fatigue and mood were not disrupted from performing this training session, with the same finding following soccer training [9]. Previous reports following competitive matches report BAM+ to be effective for monitoring readiness to train and recovery, and reduced following a single netball match [2]. Therefore, findings suggest that players may have been conditioned to this regularly performed training session, resulting in no negative effect on perceived mood or perceived fatigue. A similar

decrease in neuromuscular performance at +24h has been reported following the performance of a handball-specific training session by elite, female players [39] and following soccer training in professional male players [9]. Decreased neuromuscular function could be attributed to impaired excitation-contraction coupling resulting from low-frequency fatigue [40], with exercise-induced muscle damage and damage of type two muscle fibres [41] contributing to the decrease. Performing subsequent training in a fatigued state can impair training performance [42], adaptation to training [43] and can result in greater fatigue [6, 13]. Therefore coaches could take advantage of the recovered neuromuscular system within 2 h post-training to perform high intensity, explosive movements, rather than the following day when this type of training may be impaired, with similar observations previously reported [9, 10].

Whilst reports of elite level training intensity are limited, the demands of the present training session were similar to that of technical netball-training [6], training replicating match-play intensity [5] and to that of international-standard match-play [2]. Across an international netball tournament, heart rate (mean ± SD: 170 ± 8.7 b·min$^{-1}$), external load (8.2 ± 2.2 AU·min$^{-1}$) along with sRPE and dRPE were similar to that of the present study. Collectively, findings of the present study suggest that the training session employed successfully replicated the movement demands and internal loads of international netball match-play [2], with a similar relative intensity (training intensity compared with match-play intensity) to that previously reported across different standards of players [1, 3, 5].

We acknowledge study limitations. The training demands for different playing positions could elicit a varied response, particularly for the final part of the training session involving position-specific simulated match-play. There was no control in place for menstrual cycle phase or hormonal contraceptive use; however, hormonal responses in elite female athletes show similar patterns in response to training and competition with or without hormonal contraceptive use [12], and through the stages of the menstrual cycle [30]. Additionally, whilst no control group was present, the use of quasi-experimental designs is a good example of how an increase in external validity (i.e. the use of elite athletes) leads to a decrease in internal validity (i.e. the absence of a control group because of ethical problems with restricting a particular treatment to elite athletes) [44]. However, these are inherent limitations when conducting research in elite athletes.

## Conclusion

This is the first study to report the neuromuscular, physiological, biochemical, endocrine and perceptual responses to a netball specific session performed by professional female netball players. Primary findings indicate that the training session successfully replicated match-play intensity, responses over the 24 h period varied according to the variable being examined, disturbances were evident at 2 h post and that full recovery of all variables was not achieved within 24 h. These data indicate that the residual effects of the prior training bout should be considered when planning subsequent training within 24 h; findings likely of interest to staff involved in the planning of training for female netball players.

## Acknowledgments

We would like to acknowledge the Celtic Dragons netball team players, coaching and support staff for their support through this testing. Additionally, to sport science staff at Sport Wales for assisting with data collection.

## Author Contributions

**Conceptualization:** Laurence P. Birdsey, Christian J. Cook, Liam P. Kilduff.

**Data curation:** Laurence P. Birdsey, Liam P. Kilduff.

**Formal analysis:** Laurence P. Birdsey, Matthew Weston, Mark Russell, Liam P. Kilduff.

**Investigation:** Laurence P. Birdsey.

**Methodology:** Laurence P. Birdsey, Mark Russell, Michael Johnston, Liam P. Kilduff.

**Supervision:** Matthew Weston, Mark Russell, Christian J. Cook, Liam P. Kilduff.

**Writing – original draft:** Laurence P. Birdsey, Matthew Weston, Mark Russell, Michael Johnston, Christian J. Cook, Liam P. Kilduff.

**Writing – review & editing:** Laurence P. Birdsey, Matthew Weston, Mark Russell, Michael Johnston, Christian J. Cook, Liam P. Kilduff.

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
