## [Decision Letter · Decision Letter 0]

22 Oct 2021

PONE-D-21-16797Acute physiological and perceptual responses to a netball specific training session in professional female netball playersPLOS ONE

Dear Dr. Kilduff,

Thank you for submitting your manuscript to PLOS ONE. After careful consideration, we feel that it has merit but does not fully meet PLOS ONE’s publication criteria as it currently stands. Therefore, we invite you to submit a revised version of the manuscript that addresses the points raised during the review process.

The reviewers have highlighted several consequential points which need to be addressed prior to the manuscript being suitable for publication

We look forward to receiving your revised manuscript.

Kind regards,

Chris Connaboy

Academic Editor

PLOS ONE

Journal Requirements:

Reviewers' comments:

Reviewer's Responses to Questions

**Comments to the Author**

1. Is the manuscript technically sound, and do the data support the conclusions?

Reviewer #1: Partly

Reviewer #2: Yes

Reviewer #3: Yes

2. Has the statistical analysis been performed appropriately and rigorously? 

Reviewer #1: No

Reviewer #2: Yes

Reviewer #3: Yes

3. Have the authors made all data underlying the findings in their manuscript fully available?

Reviewer #1: Yes

Reviewer #2: No

Reviewer #3: Yes

4. Is the manuscript presented in an intelligible fashion and written in standard English?

Reviewer #1: Yes

Reviewer #2: Yes

Reviewer #3: Yes

5. Review Comments to the Author

Reviewer #1: General Comments

Thank-you for the opportunity to review your work. I enjoyed reading the manuscript and congratulate the authors on conducting this type of work in a real world high performance environment. This is not easy to do and is a great strength of the work because it maximises the applicability of the findings.

The work is very clear, concise and well written. I've made some specific comments/suggestions below for the authors to consider, some of which are probably more important than others. I fully accept that there are multiple ways to do things, and the authors have clearly attempted to justify the approaches taken. My comments are obviously the view of a first time reader so I hope they might be of some use from that perspective.

Specific Comments

L30: What exact time does +0h refer to? I assume it refers to immediately post training but providing being specific would help the reader.

L33: did you consider some threshold for the smallest important effect? A clear "difference" is one thing, but it's practical relevance is something else. You probably deal with this in detail in the methods section, and I know abstract word counts are a limitation, but a little more detail here might be useful.

Introduction: this section is very nicely written and easy to read. It does a good job of highlighting the gaps in the research and the purpose of the study.

Design

L115: given multiple papers have suggested that jump output metrics like height (and potentially power) are not impacted by team sport performance, why did you select these variables? Other metrics reflecting change in movement strategy are likely to provide more insight and these are easily calculated or automatically available in the force plate software.

L118: it may be worth justifying +24h as the final measurement point. Was there a particular reason for this? It may well limit the understanding of the extent of the post-training response.

L225: is mean HR a valid representation during intermittent activity such as Netball? Did you consider applying some kind of TRIMP?

L246: whilst I completely understand the thinking for your proposed method of determining a clear difference (which absolutely has merit), it doesn't appear to take into account what might be considered the smallest important effect for each variable. In order to interpret the meaningfulness of any clear difference, which seems to be key when considering the practical implications, this could be considered. Approaches such as a minimal effects test allow this but I understand could be limited for this type of work as it relies on interpretation of a p value. Other options might also be worth considering such as calculation of the Second Generation p Value (Blume et. al. 2018, Blume et. al. 2019 et. al.). Given you have also calculated the standardised effect, you might consider using established thresholds for small, medium, large etc.

An important issue to consider regarding the statistical analysis is the issue of dose-response. Specifically, how does the training dose interact with the size of the pre-post training change? This kind of approach would substantially increase the implications for the work.

Figures: figures are often down to personal preference in my view they don't fully convey the outcomes. Without some indication of what represents the smallest important effect, the change score +/- 95% on it's own without some reference point aren't as informative as they could be. The inclusion of the 95% CI is a nice representation of the uncertainty but perhaps consider the issue of the smallest important effect within the overall analysis approach. Granted, the figures convey the change and uncertainty relative to zero.

Discussion

L315: your finding that some markers had not returned to baseline at 24h post suggests measuring the response beyond this time would have been valuable. Was there a reason this wasn't done? I assume because normal training prevented it?

L317: why does the fact that some variables didn't return to baseline suggest training should be modulated to account for residual effects of the previous training bout? Should always be conducted in an "optimal state"? You suggest some reasons later on (L381) why training in a fatigued state may be problematic but there may well be situations where you deliberately train in fatigued state (e.g. for technical, physical and psychological reasons). Our field seems to have gone too far down the path of avoiding training so these other aspects may be worth of mention. In addition, without consideration of the dose-response aspect mentioned earlier, any changes could be unimportant.

This section is also well written. It flows really well and like the rest of the work is easy to read. Based on your approach the points are appropriate, however I feel that the insight provided is somewhat limited by the depth of analysis. As a result, I think the impact of the work on practice could actually be higher.

Reviewer #2: Dear editor and authors

Thank you for allowing me to give my opinion on this interesting paper.

Overall, the paper is well written and presents some interesting finding about the acute biochemical, physical and perceptual responses to a single training session.

Main strengths

- The manuscript is well written and clearly structured.

- The used methods are well described and the results are well presented.

- Working on elite/professional athletes is extremely interesting due to the paucity of literature regarding this specific population.

Main limitations

- The rational of the study is not enough convincing. What gap of the literature/knowledge is the study trying to fill?

- The study is purely descriptive.

- There was no control group, although acknowledged as a limitation, the study lacks for an essential pillar of scientific research. The main question is “the biochemical responses to a single netball training session are as follow”, but compared to what? To what magnitude/extent?

- The use of control group (professional male netball athletes, resistance or endurance training session, another team sport specific training session; e.g., football, rugby, basketball, etc.) is essential to measure the magnitude of the effect.

- There was neither randomization nor power calculation. How the authors know the required number of participants to detect a significant effect?

- There was little to no informations about the inclusion criteria.

- The study involved only netball athletes, which would limit the generalization of the current findings on other sport discipline.

Minor comments

Line 81: to effectively plan the content

For instance, the increase of testosterone and cortisol at +0h could be related to dehydration.

The composition, details, and the potential effect of the standardized meal on the subsequent biochemical results is not discussed.

The circannual (during which month) and circadian (time of the day) information of the study are not reported despite being discussed in the discussion section.

Presenting the actual data mean and standard deviation with the classical p value would be more informative, especially when the authors compared the current to former results in the discussion section.

L 328: I do not think that muscle repair would occur immediately after the exercise to be accountable for higher neuromuscular performance.

Reviewer #3: Thank you for the opportunity to review the study “Acute physiological and perceptual responses to a netball specific training session in professional female netball players.” The article is very well written, with a clear aim. The study design was well thought out and being able to complete this study in this level of athletes is commendable.

The discussion section could have some more in depth comparisons to previous studies, specifically pulling out the results / data of previous studies and comparing these to the present study. Furthermore, the discussion section would benefit from drawing some further concluding statements / ideas on the findings. Overall it is very well written and adds much needed information of the training demands and subsequent recovery in elite female netball athletes that is currently lacking so well done.

Specific Comments:

Introduction

Page 4, Line 81 – wording correction needed, change to ‘…effectively plan the content…’ currently reads as ‘…effectively the plan content…’

Methods

Design Section

Page 6, Line 119 – Consider moving the sentence ‘The above measures were repeated two (+2h) and 24 h (+24h) post-training’ to the end of the paragraph (line 125) to show the timeline of data collection more accurately.

Page 9, Line 203 – Was this ‘standardised warm-up’ the same as the warm up described prior to the training session? If not, please provide details of what was included in this warm up.

Page 9, Line 214 – Change word ‘were’ to ‘was’

6. PLOS authors have the option to publish the peer review history of their article (what does this mean?). If published, this will include your full peer review and any attached files.

Reviewer #1: No

Reviewer #2: No

Reviewer #3: No

---

## [Author Response · Author response to Decision Letter 0]

2 Dec 2021

Acute physiological and perceptual responses to a netball specific training session in professional female netball players

Dear editor,

We thank you for the opportunity to amend our manuscript in light of the expert reviewer’s comments. We feel that the readability and strength of our manuscript has improved as a consequence of doing so and wish to that the reviewers for taking the time to appraise our submission. A point-by-point response to each comment now follows…

Reviewer #1 comments 

General comments

Thank-you for the opportunity to review your work. I enjoyed reading the manuscript and congratulate the authors on conducting this type of work in a real world high performance environment. This is not easy to do and is a great strength of the work because it maximises the applicability of the findings.

The work is very clear, concise and well written. I've made some specific comments/suggestions below for the authors to consider, some of which are probably more important than others. I fully accept that there are multiple ways to do things, and the authors have clearly attempted to justify the approaches taken. My comments are obviously the view of a first time reader so I hope they might be of some use from that perspective.

Comment: Thank for you these kind words. Please see a point by point response to these comments below. As a consequence, we believe the quality of this manuscript to be enhanced, so thank you for the time taken.

Specific comments

L30: What exact time does +0h refer to? I assume it refers to immediately post training but providing being specific would help the reader.

Response: Thank you for this response. It is clear that this is not well defined for the reader. +0h, +2h, +24h refer to immediately post, two hours post and 24 hours post-training respectively.

Action: This has now been amended and reads:“…markers were assessed at baseline (immediately before; Pre), and immediately, two and 24 hours after training (+0h, +2h, +24h).

L33: did you consider some threshold for the smallest important effect? A clear "difference" is one thing, but it's practical relevance is something else. You probably deal with this in detail in the methods section, and I know abstract word counts are a limitation, but a little more detail here might be useful.

Response: Thank you for this response. In light of the reviewer’s comments, we have redrafted the figures to include a zone of what many regard as the smallest worthwhile effect (i.e., 0.2SD). We have also acknowledged this in the revised statistical methods paragraph. However, due to word limit constraints we elected not to include this in the abstract to allow more of a focus on methods and results.

Action: Figures have been redrawn and the statistical methods paragraph has been revised.

Introduction

This section is very nicely written and easy to read. It does a good job of highlighting the gaps in the research and the purpose of the study.

Response: Thank you for these kind words.

Design

L115: given multiple papers have suggested that jump output metrics like height (and potentially power) are not impacted by team sport performance, why did you select these variables? Other metrics reflecting change in movement strategy are likely to provide more insight and these are easily calculated or automatically available in the force plate software.

Response: Thank you for this response. Indeed, many variables have been identified from countermovement jump testing with reliability and response to a variety of training stimuli published (i.e. Cormack et al., 2008; Gathercole et al, 2015; Kennedy and Drake, 2021). We elected to use peak power output and jump height for several reasons. Firstly, both peak power output and jump height are sensitive to detect changes in neuromuscular performance following team-sport activities including soccer training (Sparkes et al., 2018), rugby match-play (West et al., 2014) and indeed following netball training (Birdsey et al., 2020) and netball match-play (Birdsey et al., 2019; Wood et al., 2013). Secondly, peak power output and jump height mirror changes in strategy measures (such as eccentric, concentric or total jump duration, ratio of flight time to contact time), however decrease more than these measures, and are therefore considered most sensitive to fatigue in the early stage of recovery process, such as the time-frames present in this manuscript (Kennedy and Drake, 2017). Additionally, peak power output and jump height are more reliable than strategy measures (Kennedy and Drake, 2017).

Action: This has been amended to include justification for the reader:” These measures were chosen as they are sensitive to changes following team sport training [9], netball training [6] and netball match-play [2], and can be considered most sensitive to fatigue in the early stages of the recovery process [19].”

L118: it may be worth justifying +24h as the final measurement point. Was there a particular reason for this? It may well limit the understanding of the extent of the post-training response.

Response: Thank you for this response. The final measurement of 24 h was used as this was typically the time-point when players would train again the following day when performing one training session per day. Indeed, further measures beyond this time-point would have been valuable. However, due to training commitments of these elite athletes (i.e. they undertook a training session following the measures at +24h) this was unfortunately not possible.

Action: This has now been amended to justify +24h as the final measurement point and reads: “Due to players training schedules, +24h was the final available time-point before players performed a subsequent training session” in the Design section.

L225: is mean HR a valid representation during intermittent activity such as Netball? Did you consider applying some kind of TRIMP?

Response: Thank you for this comment. Mean and maximum heart rate have been used to describe the internal intensity of netball training (Birdsey et al., 2021, Chandler et al., 2014) and competition (Birdsey et al., 2019). Birdsey et al (2019) reported lower external load (accelerometry), internal load (RPE metrics), and heart rate between when comparing goal-based to mid-court based positional groups. Mean and maximum heart rate have also been reported to differ following different types of netball training, with different activities and accumulation of external load (Chandler et al., 2014). This therefore suggests that mean and maximum HR are sensitive and related to external load (i.e. movement demands) and a valid marker of internal intensity in netball.

Action: No action required.

L246: whilst I completely understand the thinking for your proposed method of determining a clear difference (which absolutely has merit), it doesn't appear to take into account what might be considered the smallest important effect for each variable. In order to interpret the meaningfulness of any clear difference, which seems to be key when considering the practical implications, this could be considered. Approaches such as a minimal effects test allow this but I understand could be limited for this type of work as it relies on interpretation of a p value. Other options might also be worth considering such as calculation of the Second Generation p Value (Blume et. al. 2018, Blume et. al. 2019 et. al.). Given you have also calculated the standardised effect, you might consider using established thresholds for small, medium, large etc.

Response: We appreciate the reviewers comment here. While we have justified our statistical approach, we have edited slightly the statistical methods and inserted a zone denoting the smallest worthwhile effect into each figure. This way, the readers can now interpret magnitude for themselves in the way the feel comfortable in doing (i.e., raw units, standardised effect sizes, or % changes [in text only]).

Action: Statistical methods have been edited to insert a zone denoting the smallest worthwhile effect in to each figure.

An important issue to consider regarding the statistical analysis is the issue of dose-response. Specifically, how does the training dose interact with the size of the pre-post training change? This kind of approach would substantially increase the implications for the work.

Response: Thank you for this response. We attempted to characterise an array of responses following a typically performed training session, data which, up to this point has been unavailable, and feel that we have therefore added valuable information to the body of literature available in elite female athletes. Answering the suggested question is an important next step, and something that other researchers will hopefully design studies specifically for to build upon this work. However, this was not an aim of this study, and as this was not specifically hypothesised in advance, caution about any such relations would have been required.

Action: We thank you for this comment, however no action taken.

Figures: figures are often down to personal preference in my view they don't fully convey the outcomes. Without some indication of what represents the smallest important effect, the change score +/- 95% on it's own without some reference point aren't as informative as they could be. The inclusion of the 95% CI is a nice representation of the uncertainty but perhaps consider the issue of the smallest important effect within the overall analysis approach. Granted, the figures convey the change and uncertainty relative to zero.

Response: Thank you for this response. In light of the reviewer’s comments, we have redrafted the figures to include a zone of what many regard as the smallest worthwhile effect (i.e., 0.2SD). We have also acknowledged this in the revised statistical methods paragraph.

Action: Figures have been redrafted with a zone of what many regard as the smallest worthwhile effect. Statistical analyses section now includes: “Further, a region denoting 0.2*SD, commonly referred to as a smallest worthwhile effect, has been included in all forest plots (see Figures 1-3).”

Discussion

L315: your finding that some markers had not returned to baseline at 24h post suggests measuring the response beyond this time would have been valuable. Was there a reason this wasn't done? I assume because normal training prevented it?

Response: Thank you for this response. Indeed, as all measures had not returned to baseline at +24h, and no further measures were made, it is impossible to determine when full recovery would have occurred. However, that is correct, due to the nature of the elite player’s training schedule, at this time-point players performed a further training session.

Action: This information has now been included for the reader: “As players performed a subsequent training session at this time-point as part of their team’s elite training schedule, no further measures could be obtained”.

L317: why does the fact that some variables didn't return to baseline suggest training should be modulated to account for residual effects of the previous training bout? Should always be conducted in an "optimal state"? You suggest some reasons later on (L381) why training in a fatigued state may be problematic but there may well be situations where you deliberately train in fatigued state (e.g. for technical, physical and psychological reasons). Our field seems to have gone too far down the path of avoiding training so these other aspects may be worth of mention. In addition, without consideration of the dose-response aspect mentioned earlier, any changes could be unimportant.

Response: Thank you for this response. We fully acknowledge that at times, overload and fatigue may be desired and even necessary for specific adaptations. We did not intend to suggest that our findings suggest that training must always be adapted. Instead, we try to highlight that the residual effects of the prior training stimulus may have an impact on subsequent training, therefore should be considered by the coaching staff.

Action: This has been amended to make this clearer for the reader: “Accordingly, these data indicate that the residual effects of the previous training bout should be considered when planning subsequent training in the 24 h following a netball training session; findings which will likely be of interest to staff involved in the planning and periodisation of training for female netball players.

This section is also well written. It flows really well and like the rest of the work is easy to read. Based on your approach the points are appropriate, however I feel that the insight provided is somewhat limited by the depth of analysis. As a result, I think the impact of the work on practice could actually be higher.

Response: Thank you for these kind words. We feel that owing to these responses from the reviewer, our manuscript is better off for it.

Reviewer #2 comments 

Dear editor and authors

Thank you for allowing me to give my opinion on this interesting paper. Overall, the paper is well written and presents some interesting finding about the acute biochemical, physical and perceptual responses to a single training session.

Main strengths

- The manuscript is well written and clearly structured.

- The used methods are well described, and the results are well presented.

- Working on elite/professional athletes is extremely interesting due to the paucity of literature regarding this specific population.

Comment: Thank for you these kind words. Please see a point by point response to these comments below. As a consequence, we believe the quality of this manuscript to be enhanced, so thank you for the time taken.

Main limitations

- The rational of the study is not enough convincing. What gap of the literature/knowledge is the study trying to fill?

Thank you for this suggestion. It is clear that the rationale for this study was not justified sufficiently.

Action: This has been amended and now reads: “While netball-specific training responses are scarce, the acute post-exercise responses to training in other sports have been extensively reported following isolated strength [7], endurance [8] and soccer [9] training, with a single observation following speed training [10]; all of which have application to the demands of netball players. However, as players perform training to improve aspects related specifically to match performance, sport-specific training sessions are key to fully understand the responses of netball-specific training. Following soccer team-sport training, immediate increases in testosterone and decreases in cortisol concentrations have been observed in addition to a bi-modal recovery pattern of neuromuscular performance, with an initial decrease immediately post, partial recovery at two, and further impairment at 24 h post [9]. However, in female players, a delayed endocrine response has been reported of 24 h, with responses evident up to 72 h post-training [11], whilst following field hockey training, exercise intensity influences the endocrine response [12]. A greater understanding of the acute responses to, and recovery profile from on-court netball training may assist coaches and conditioning coaches to effectively plan the content of individual sessions, as well as the positioning of training within the week.”

At present, there are limited reports upon the acute responses to team-sport training in females, with only one report following netball-specific training [6]. Knowledge of both the training stimulus, as well as the recovery response are necessary to prevent cumulative fatigue [6,13] and allow recovery for adaptation. It is therefore imperative that coaching and conditioning staff have an understanding of the acute responses to specific training sessions to assist with effectively planning and optimising training.”

- The study is purely descriptive.

Response: Thank you for this observation. Indeed, as this is the first study of its kind in elite female netball, we elected to characterise responses in an attempt to better understand what the demands and consequences were to this type of training session. There are very few reports in an elite female population, especially in netball, making it challenging for staff to understand how they may balance training and recovery, or how to implement recovery strategies. We hope this characterisation supports future studies investigating responses to a wider range of training types, adaptation to training and perhaps recovery interventions.

Action: None required

- There was no control group, although acknowledged as a limitation, the study lacks for an essential pillar of scientific research. The main question is “the biochemical responses to a single netball training session are as follow”, but compared to what? To what magnitude/extent?

Response: Thank you for this observation. Quasi-experimental designs are possible (e.g. Partickand Hrycaiko, 1998) in which all elite athletes in the study are exposed to treatment and control periods sandwiched between multiple observations of performance over time. The use of quasi-experimental designs is a good example of how an increase in external validity (i.e. the use of elite athletes) leads to a decrease in internal validity (i.e. the absence of a control group because of ethical problems with restricting a particular treatment to elite athletes) (Atkinson, G. & Nevill, A. M. Selected issues in the design and analysis of sport performance research. Journal of Sports Sciences 19, 811–827 (2001).

Action: No action required

- The use of control group (professional male netball athletes, resistance or endurance training session, another team sport specific training session; e.g., football, rugby, basketball, etc.) is essential to measure the magnitude of the effect.

Response: Response: Please see our previous response.

Action: No action required

- There was neither randomization nor power calculation. How the authors know the required number of participants to detect a significant effect?

Response: Thank you for this observation. As we performed an exploratory study, there can be little value of a power calculation where scares data are available on which to base the calculations (Jones et al., 2003). We therefore elected not to perform a power calculation.

Action: We have revised the statistical methods to include a justification for not performing a power analysis. This reads: “We elected not to perform a power calculation as they are of little value in early exploratory studies, such as ours, where scarce data are available on which to base the calculations (Jones et al., 2003).” 

- There was little to no informations about the inclusion criteria.

Response: Thank you for this response. This was an oversight, and it is clear that this information is lacking for the reader. 

Action: This has now been included: “Players were included as members of this professional netball team and determined to be available for training by the team physiotherapist.”

- The study involved only netball athletes, which would limit the generalization of the current findings on other sport discipline.

Response: Thank you for this response. Indeed, responses to training are specific to the exercise mode, intensity as well as population studied. This is one of the main reasons for conducting this research, especially in elite populations where responses can be different to that of un-trained. However, particularly as there is such a difference in volume of reports in high-standard female sport compared to that of male, there is information within this manuscript which the authors feel is of value for coaches and conditioning coaches in different sports. 

Action: No action required.

Minor comments

Line 81: to effectively plan the content

Response: Thank you for this response,

Action: This has now been amended to include “the”

For instance, the increase of testosterone and cortisol at +0h could be related to dehydration.

Response: Thank you for this response. Indeed, evidence suggests that dehydration can affect salivary hormone concentrations. However, this is unlikely to have been a contributing factor in the present study as players arrived to training hydrated, as per the recommendation of the team nutritionist, which was commonly performed for all training. Additionally, players could consume fluid during the training session ad libitum during breaks in play to avoid dehydration, and indeed body mass measured as part of countermovement jump testing indicates no change in mass from pre- to post-training.

Action: No action required.

The composition, details, and the potential effect of the standardized meal on the subsequent biochemical results is not discussed.

Response: Thank you for this response. This detail was not provided and was an oversight by the authors.

Action: This has been amended to include this detail:”In preparation for training, players were instructed to eat and drink as usual (i.e. a high carbohydrate meal to support carbohydrate availability for the training session) and consumed a standardised meal prescribed by the team nutritionist to support recovery (i.e. high in carbohydrates to replenish carbohydrate stores, in protein to support muscle protein resynthesis, and with fruit and vegetables as part of a balanced diet) immediately following the measurements collected post-session at +0h.”

The circannual (during which month) and circadian (time of the day) information of the study are not reported despite being discussed in the discussion section.

Response: Thank you for this suggestion. This was not included, it is clear that this information is lacking for the reader.

Action: This has been amended and now reads: “…were recruited for this study that was conducted in December during the 2016 pre-season period…”

“This observational study was conducted over a 24 h period that followed an on-court netball-specific training session commencing at 16:00 h”

Presenting the actual data mean and standard deviation with the classical p value would be more informative, especially when the authors compared the current to former results in the discussion section.

Response: Thank you for this comment. As we directly tested and estimated the differences between time points, it can be noted that if the two 95 % confidence intervals fail to overlap, then when using the same assumptions used to compute the confidence intervals, as we can, we will find P < 0.05 for the difference; and if one of the 95 % intervals contains the point estimate from the other group or study, we will find P > 0.05 for the difference (Greenland et al., 2016). As such, we elected not to report p values as this would not provide any additional information that what is already presented. Greenland, S. et al. Statistical tests, P values, confidence intervals, and power: a guide to misinterpretations. European Journal of Epidemiology 31, 337–350 (2016).

Action: No action required.

L 328: I do not think that muscle repair would occur immediately after the exercise to be accountable for higher neuromuscular performance.

Response: Thank you for this suggestion. This line is not referring to neuromuscular performance, rather hormonal responses, particularly an increase in testosterone and cortisol concentrations at this time-point. However, clearly this is not easily followed and has led to confusion.

Action: This has been amended and now reads: “Increases in testosterone and cortisol concentrations could be attributed to an increase in competitiveness and dominance behaviours [25,26], as well as energy provision and muscle tissue repair following exercise-induced muscle damage [27].”

Reviewer #3 comments 

Thank you for the opportunity to review the study “Acute physiological and perceptual responses to a netball specific training session in professional female netball players.” The article is very well written, with a clear aim. The study design was well thought out and being able to complete this study in this level of athletes is commendable.

Comment: Thank for you these kind words. Please see a point by point response to these comments below. As a consequence, we believe the quality of this manuscript to be enhanced, so thank you for the time taken.

The discussion section could have some more in depth comparisons to previous studies, specifically pulling out the results / data of previous studies and comparing these to the present study. Furthermore, the discussion section would benefit from drawing some further concluding statements / ideas on the findings. Overall it is very well written and adds much needed information of the training demands and subsequent recovery in elite female netball athletes that is currently lacking so well done.

Response: Thank you for this suggestion and we have attempted to improve aspects of this in line with your comment.

Specific Comments

Introduction

Page 4, Line 81 – wording correction needed, change to ‘…effectively plan the content…’ currently reads as ‘…effectively the plan content…’

Response: Thank you for this suggestion and finding this error.

Action: This has been amended and now reads: “to effectively plan the content”

Methods

Design Section

Page 6, Line 119 – Consider moving the sentence ‘The above measures were repeated two (+2h) and 24 h (+24h) post-training’ to the end of the paragraph (line 125) to show the timeline of data collection more accurately.

Response: Thank you for this suggestion. This flows better as a consequence, introducing +0h first.

Action: Amended and now reads: “Measures recorded immediately prior to the training session were repeated two (+2h) and 24 h (+24h) post-training.”

Page 9, Line 203 – Was this ‘standardised warm-up’ the same as the warm up described prior to the training session? If not, please provide details of what was included in this warm up.

Response: Thank you for this comment. That is correct, the same standardised warm-up was used throughout testing. The only difference was immediately post-training, when players performed two practice jumps only.

Action: No action required

Page 9, Line 214 – Change word ‘were’ to ‘was’

Response: Thank you for this suggestion.

Action: This has been amended as suggested.

---

## [Decision Letter · Decision Letter 1]

27 Jan 2022

Acute physiological and perceptual responses to a netball specific training session in professional female netball players

PONE-D-21-16797R1

Dear Dr. Kilduff,

We’re pleased to inform you that your manuscript has been judged scientifically suitable for publication and will be formally accepted for publication once it meets all outstanding technical requirements.

Kind regards,

Chris Connaboy

Academic Editor

PLOS ONE

Additional Editor Comments (optional):

Reviewers' comments:

Reviewer's Responses to Questions

**Comments to the Author**

1. If the authors have adequately addressed your comments raised in a previous round of review and you feel that this manuscript is now acceptable for publication, you may indicate that here to bypass the “Comments to the Author” section, enter your conflict of interest statement in the “Confidential to Editor” section, and submit your "Accept" recommendation.

Reviewer #1: All comments have been addressed

Reviewer #3: All comments have been addressed

2. Is the manuscript technically sound, and do the data support the conclusions?

Reviewer #1: Yes

Reviewer #3: Yes

3. Has the statistical analysis been performed appropriately and rigorously? 

Reviewer #1: Yes

Reviewer #3: Yes

4. Have the authors made all data underlying the findings in their manuscript fully available?

Reviewer #1: Yes

Reviewer #3: Yes

5. Is the manuscript presented in an intelligible fashion and written in standard English?

Reviewer #1: Yes

Reviewer #3: Yes

6. Review Comments to the Author

Reviewer #1: Thank-you for addressing my original comments. In my view, the majority of responses are adequate although I'm still not convinced by the choice of jump variables and the justification for the choice.

Reviewer #3: (No Response)

7. PLOS authors have the option to publish the peer review history of their article (what does this mean?). If published, this will include your full peer review and any attached files.

Reviewer #1: No

Reviewer #3: No

---

## [Editor Report · Acceptance letter]

31 Jan 2022

PONE-D-21-16797R1 

Acute physiological and perceptual responses to a netball specific training session in professional female netball players 

Dear Dr. Kilduff:

I'm pleased to inform you that your manuscript has been deemed suitable for publication in PLOS ONE. Congratulations! Your manuscript is now with our production department. 

Kind regards, 

on behalf of

Dr. Chris Connaboy 

Academic Editor

PLOS ONE